# How Can You Know the Bible and Not Believe in Our Lord? Guiding Pilgrims across the Jewish–Christian Divide

Jackie Feldman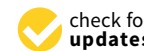

Department of Sociology and Anthropology, Ben Gurion University of the Negev, 8410501 Be'er-Sheva, Israel; jfeldman@bgu.ac.il

**Abstract:** Drawing on auto-ethnographic descriptions from four decades of my own work as a Jewish guide for Christian Holy Land pilgrims, I examine how overlapping faiths are expressed in guide–group exchanges at Biblical sites on Evangelical pilgrimages. I outline several faith interactions: Between reading the Bible as an affirmation of Christian faith or as a legitimation of Israeli heritage, between commitments to missionary Evangelical Christianity and to Judaism, between Evangelical practice and those of other Christian groups at holy sites, and between faith-based certainties and scientific skepticism. These encounters are both limited and enabled by the frames of the pilgrimage: The environmental bubble of the guided tour, the Christian orientations and activities in the itinerary, and the power relations of hosts and guests. Yet, unplanned encounters with religious others in the charged Biblical landscape offer new opportunities for reflection on previously held truths and commitments. I conclude by suggesting that Holy Land guided pilgrimages may broaden religious horizons by offering an interreligious model of faith experience based on encounters with the other.

**Keywords:** tour guiding; interreligious dialogue; evangelicals; Judaism; Holy Land; pilgrimage; Israel

## 1. Introduction

At the end of a long day touring the Galilee, Pastor Larry gathered his flock at the Mount of Beatitudes; drawing on Jesus' Sermon on the Mount, he preached:

You have heard it said: "Imprint many images on your digital memory cards";

but I say: "Imprint the image of your father on your heart and mind".

You have heard it said: "This is the trip of a lifetime";

but I say: "I am your God for all time".

You have heard it said: "The historical perspective is highly educational";

but I say: "Learn of Me".

You have heard it said: "Carry your snacks, water, hats, coats, suntan lotion";

but I say: "Eat of me and I am your covering".

You have heard it said: "The water level has changed, the natural borders have changed, the

rainfall has changed";

but I say: "I never change"[2].

---

[2]    Whether the Sermon on the Mount and, in general, Matthew's Gospel advocate the abrogation of the Old Testament Law in the Christian community is a matter of contemporary scholarly debate (for a summary, see Deines 2008). What is important here is that most Evangelical Christians, like Pastor Larry, believe that this is so.

The sermon Pastor Larry preaches draws on a canonical New Testament text (Mt. 5), allegedly spoken by Jesus at the site, which builds on the Ten Commandments. It contrasts the old dispensation with the new, the requirements of the Law with the prescriptions of faith, the external with the internal, and the Pharisees and scribes with the Christian Messiah.[2] The pastor adapts Jesus' words in the Sermon on the Mount to contrast the narratives and habitus of tourism, as well as the mediation of objects and landscapes—as represented by the Jewish-Israeli tour guide; with the spiritual, supra-material message of Christianity—as spoken by the pastor. I have worked as a Jewish guide for Christian pilgrims to the Holy Land for nearly four decades. My Jewish background and my desire, as an immigrant, to manifest my attachment to Israel were major influences bringing me to seek work as a pilgrim guide (Feldman 2018). While my Jewish knowledge and belonging endowed me with a certain status as a member of the "people of the Book", insofar as it validated the shared background of Christians and Jews, the differences between Christian understandings of Jews and my Jewish self-understanding engendered tensions that needed to be skillfully negotiated. After over two decades of work guiding, with several hundred pilgrim groups of a variety of denominations from the USA, UK, Germany, the Netherlands, France, Nigeria, Belgium, and India, I began to critically analyze my experiences as a tour guide. I then supplemented them with interviews and conversations with other guides, group leaders, and pilgrims to confirm or question how typical my own experiences were for guide–pilgrim interactions as a whole (for accounts of other guides, see Feldman 2015). The descriptions of Holy Land pilgrimage encounters provided in this article are based primarily on auto-ethnography of my experiences with English-speaking Evangelicals. While this auto-ethnography focuses on interactions with groups on one- to two-week-long package trips, this is representative of a majority of Holy Land pilgrimages.

Auto-ethnography has been employed increasingly as a research tool over the past several decades, especially since the narrative and reflexive turns in anthropology (Clifford and Marcus 1986) emphasized the subjectivity and positionality of the researcher. It was one of several new ways of presenting the field, "bringing out the experiential, interpretive, dialogical, and polyphonic process at work in . . . ethnography" (Marcus 2007, p. 1128.) The rigor of an auto-ethnography is summarized by Le Roux as encompassing five characteristics: 1. Subjectivity: The visible self-conscious involvement of the researcher/author in the construction of the narrative which constitutes the research. 2. Self-reflexivity: The researcher's intense awareness of his or her role in the research, which is situated within a historical and cultural context. 3. Resonance: The engagement of the audience with the writer's story on an intellectual and emotional level. 4. Credibility: The evidence of verisimilitude, plausibility, and trustworthiness in the research. 5. Contribution: The capacity of the work to extend knowledge, generate ongoing research, liberate, empower, improve practice, or make a contribution to social change (Le Roux 2017, p. 204).

Auto-ethnography is particularly well suited to describe the subtleties of verbal and symbolic interaction that occur within the frame of the guided pilgrimage. It enables me to focus on the roles and positions assumed when competing but intertwined faiths and practices interact. It also enables a dialogical presentation of the definition of a key term in pilgrimage in general and in this special issue in particular—'faith'. Faith means different things, plays different roles, and is made manifest in different forms among the various actors in the field. The self-reflection required in auto-ethnography presents these differences and the emotional resonances they engender. Auto-ethnographic description and subsequent reflection demonstrate how the meanings of pilgrimage, for both pilgrims and locals involved with them, are not predetermined, but develop through performance. My own faith trajectory, as a once-Orthodox Jew and committed but increasingly critical Israeli, expressed itself and was, in turn, shaped by presenting Judaism, Israel, and Christianity to a Christian public.

---

[2]  Whether the Sermon on the Mount and, in general, Matthew's Gospel advocate the abrogation of the Old Testament Law in the Christian community is a matter of contemporary scholarly debate (for a summary, see Deines 2008). What is important here is that most Evangelical Christians, like Pastor Larry, believe that this is so.

I begin with an auto-ethnographic introduction of my background with respect to faith, praxis, and doubt, and continue with a description of the logistic and social frames of the Christian pilgrimages, and the training and ideological orientations of Jewish-Israeli tour guides. I then outline several intersections of conflicting and complimentary faith that surface in guiding: Between reading the Bible as an affirmation of Christian faith or as a legitimation of Israeli heritage, between commitments to missionary Evangelical Christianity and to Judaism, between Evangelical practice and those of other Christian groups at holy sites, and between faith-based certainties and scientific skepticism. I conclude by suggesting that Holy Land guided pilgrimages may offer a model of faith experience based on encounters with the other.

The examples from the Holy Land provided below shed light on broader issues in the border zones[3] (Bruner 2004, p. 17) of hosts and guests (Candea and da Col 2012), of pilgrimage and heritage tourism, and of interreligious dialogue and ritual (Moyaert 2019). They also explore the capacity and limits of cultural mediators, such as tour guides, in accommodating or challenging the views of visitors. The potential encounters are both circumscribed and enabled by the frames of the pilgrimage: The environmental bubble of the guided tour, the Christian orientations and activities in the itinerary, and the power relations of hosts and guests. Yet, unforeseen encounters with religious others in the charged Biblical landscape offer new opportunities for reflection on previously held truths and commitments.

## 2. Faith, Pilgrimage, and Practice: An Auto-Ethnographic Prelude

In the study of other societies, we often ask, "what do they believe?" as if, once that formulation is achieved, we can identify the nature of the group and its boundaries and explain individual actions. Yet belief itself is extremely fluid, as are the relationships between belief, practice, belonging, and identity. Ruel (1997) has summarized how in the history of Christianity, belief or faith[4] has taken on a wide variety of meanings. From Abraham's trust in his relationship with God, to Paul's emphasis on the conversion experience as the foundation of Christian faith and marker of group adherence, to the Nicene faith in a particular dogma as truth and an identifying sign of belonging (Ruel 1997, p. 46), to Luther's location of God's grace in the conviction that results from an inward struggle. The emphasis on belief, Ruel shows, is particular to modern forms of Protestantism, and in placing this question at center, we subject other societies and religions to often inappropriate Protestant frameworks.

The issue surfaced when the Protestant pilgrims I guided not infrequently asked me: "Are you a believing Jew?" It took me a while to understand the question. Let me answer with a story (is that not what Jesus would do?):

> A high school rabbi calls his pupil Yankel in for a talk.
>
> "Yankel", he asks, "I didn't see you at morning prayers for the last two weeks. Where have you been?"
>
> "Rebbi", he replies, "I wanted to talk to you about that. Lately, I've been having doubts".
>
> "What do you mean 'doubts'?"
>
> "Well, rebbi", he stutters, "it's like ... I'm not sure God exists".
>
> "But who *asked* you that!".

As an Orthodox Jewish kid, growing up in New York, what we believed in was rarely discussed, even in the 'sacred studies', which took up several hours of our daily education. What was important

---

[3]   I rely here on Bruner's definition of border zones in the context of tourism: "distinct meeting places between the tourists who come forth from their hotels and the local performers, the 'natives', who leave their homes to engage the tourists in structured ways in predetermined localities for defined periods of time" (Bruner 2004, p. 17).

[4]   Ruel declares that there is "a sufficient continuity and overlap in meaning between 'faith' and 'belief' to allow 'belief' to do duty for both" (Ruel 1997, p. 37).

was what we did. The borders were clearly marked as 'Jew' and 'Gentile'; the language we spoke, the holidays we celebrated, the food we ate, the side of Broadway we walked on—were far more significant than any affirmation of belief.

One of my motivations for leaving New York and immigrating to Jerusalem was to escape the confines of Jewish Orthodoxy. The Land of Israel was for me a place of wide-open spaces and a Jewish majority that did not need to enclose itself in a ghetto or accommodate itself to majority Gentile (and sometimes implicitly Christian) expectations. The Land, the Hebrew language, and the Jewish majority living there would help me develop a critical Zionist identity open to the world—one that did not need Orthodox practice to sustain it. In this sense, my choice to move to Israel and to learn the lay of the land and its Biblical geography and history through the tour guide course was an affirmation of faith and belonging. Even my interest in the New Testament was furthered by the discovery of a Jewish Jesus who could overturn the tables of the money-changers, and challenge the legal hair-splitting of the Pharisees, in whom I recognized my childhood rabbis. The Jesus of the New Testament attracted me as a rebel, not as a man of faith.

So, does that make me a "believing Jew?"

In the course of time, I came to understand that the pilgrim's question meant: "As a Jew, do you believe in Jesus Christ as your Lord and Savior"?

Now, the answer was clear to me: "No"—meaning, "I don't belong to your religious community".[5] The answer to the pilgrim, if only for diplomatic reasons, might be longer.

## 3. Seekers of Jesus Meet Ambassadors of Israel: The Guided Pilgrimage Frame

In 2018, of the 4.1 million incoming tourists to Israel, 55% identified as Christians, 24% as pilgrims. Among the latter, 43% were Catholic, 31% Protestant (83% of them, Evangelical), and 24% Orthodox (Israel Ministry of Tourism 2019, p. 10)[6]. The majority arrived in organized groups, mostly from a single church or parish. The groups' seven- to ten-day itineraries focus on sites of significance to Christian faith and history, and are frequently advertised as "a walk in the footsteps of Jesus." Pilgrims regularly conduct Christian worship, read Bible passages, and sing hymns during their visit. Although they share a common faith, drawing them to the sites of Jesus' birth, baptism, preaching, Passion, crucifixion, and resurrection, their itineraries, worship practices, and understandings differ substantially from one denomination to another (Bowman 1991).

The focus of almost all pilgrimages is on New Testament and Old Testament sites. The itineraries privilege enclaval spaces (Edensor 2000, pp. 332–33), including the chapels, hotels, and pilgrim hostels to which pilgrims retreat each evening. The tour bus isolates the group from surrounding auditory and olfactory input, keeping the Orient safely on the other side of the window,[7] while intensifying the social interaction within the bubble. Most schedules are densely packed, with little free time. While a minority of tours include meetings with local inhabitants, often in order to express solidarity with Palestinians/Palestinian Christians or Israeli settlers (Feldman 2011), for most, the tour guide (and sometimes, the bus driver) is the only local person they converse with in the course of their visit.

---

5　My personal position was well expressed by R.B. Braithwate, for whom religious assertions are primarily declarations of adherence to a policy of action, declarations of commitment to a way of life. As Gellman summarizes, according to Braithwaite, then, "to say that 'God exists' or that 'God sanctions the Jewish religion' would not be to assert 'a truth', but to express a positive attitude towards a way of life associated with those sentences, or to express a commitment to an associated way of life . . . What distinguishes a religious Jew from a Christian are the dissimilar stories they tell in their respective ways of life" (Braithwaite 1971, pp. 78–86 in Gellman 2020, p. 37). My position, as opposed to that of many of the pilgrims (and Gellman himself), is not made explicit in the guiding interactions; it is not a philosophical discussion.

6　I thank Noga Collins-Kreiner for sharing the Ministry of Tourism's annual reports with me.

7　There is a substantial literature on the links between Protestant Holy Land pilgrimage, Biblical archaeology, Orientalism, and the colonial project beginning with the mid-late 19th-century. Thomas Cook, perhaps the founder of the package tour industry, offered individual bathing huts at the Jordan River so that British pilgrims could avoid contact with the Russian peasants who were the largest group of pilgrims at the time. See (Feldman and Ron 2011; Lock 2003; Rogers 2003).

　　　Some groups, primarily Orthodox and more conservative Catholics, prefer to be guided by monks or priests based in Jerusalem or in their home countries. Most, however, engage the services of a Palestinian (Christian or Muslim) or Jewish-Israeli guide to work along with the priest or pastor who accompanies the group from home. Local (Palestinian or Israeli) travel agents are aware of the religious differences. In reply to a request that he supply guides whose faith and worship experience were similar to those of the Evangelical pilgrims, the travel agent I most often worked for replied: "It is not a question of guides believing in the inerrancy of the Bible. It is whether the guides are sensitive enough to the belief systems of their Evangelical audiences so as not to create doubts about the authenticity of the Bible within their guiding expositions"[8] (e-mail correspondence, January 2012).

　　　As Evangelicals see present-day (Israeli) Jews as People of the Book, representatives of Biblical Israel, bearers of the promise, true natives of the land, and future witnesses to Jesus' Second Coming, they will often request (and be granted) Jewish guides. Both the roles of guides on package tours and historical Christian attitudes towards Judaism place them in the position of mediators between Christian pilgrims and their sacra—the Bible, Jesus, and the holy places. This role grants the Jewish-Israeli guide status and authority. It is also a source of role strain (Hochschild 1983; Goffman 2005), insofar as the group's expectations diverge from guides' own self-understandings as Jews and Israelis.

　　　Israelis choose to become tour guides for a variety of reasons. A large portion, perhaps half of active licensed Israeli tour guides, are immigrants from other countries. For many of them, as for me, participation in the two-year-long course, learning history, botany, and archaeology and participating in seventy-five full-day hikes, is an initiation into Israeliness (Feldman 2018). Knowledge of the Land though hiking—*yedi'at ha'aretz*—is important in Israeli education and military service (Selwyn 1996; Almog 2000, pp. 160–84), and in the tour guide course. Israeli belonging is depicted as an act of *recovery* of Biblical roots detached at the time of the destruction of the Second Temple and the beginning of exile (Zerubavel 1995). The Zionist reading of the Hebrew Bible (Old Testament), especially in situ, enacts it as a national history book, source of heritage, and justification of the claim to the land.

　　　As David Lowenthal formulated it, consuming secular heritage is a kind of religious experience and should be conceptualized in religious terms. In his view, the secular legacy not only superseded religious heritage, but has become a religion or a cult in and of itself. It played a decisive role in the production or preservation of memory of a personal, collective, or national community—one which arouses strong emotions. Heritage often tolerates no doubts or compromise. Like any faith, it also provides a dose of enchantment, and a tendency towards the emotional and the devotional (Lowenthal 1996, p. 2). Hence, places that are both sites of heritagization and religious devotion may be conceptualized as shared shrines or as contested sanctuaries (Bear et al. 2020).

　　　Israeli hikes may be seen as a type of secular pilgrimage, in which "information and interpretations are selected primarily in order to arouse feelings of belonging to the place and to evoke collective experience of identification with symbolic heroes, groups, and localities" (Katz 1985, p. 63). In the formation of the Israeli teacher–guide, archeological relics or historical events are intentionally loaded with national meanings. At historical and archaeological sites, "original time tables are contracted ('shrunk') and original orders and causal links are modified. It is done in order to dramatize the scholarly 'stories' and consequently to give the teacher–guide audience the feeling of witnessing scenes and heroes of the past, as if they were taking place here and now" (Katz 1985). The Israeli teacher–guide, in Hebrew, *moreh derekh*—teacher of the way, "is more than a 'teacher of the way'; . . . he is an encourager of faith" (69). There is much common ground between the pilgrims' quest to make the Bible real, confirm their faith, walk in the footsteps of Jesus, the prophets, and ancestors; the immigrant guide's quest to become an Israeli native, and the native Israeli's desire to root himself in the land through "conquering it with one's feet" (Katriel 1995).

---

[8]　Some travel agents catering to the Evangelical market emphasize their Messianic orientation, including that of their guides, as a selling point.

Tour guides, in general, are "entrusted with the public relations mission to encapsulate the essence of place . . . and to be a window onto a site, region, or even country"[9] (Salazar 2005, p. 629). An Israel Ministry of Tourism official overseeing the tour guide course said: "The guide is the face of the country, and leaves an impression on tourists . . . the course also gives messages linked to the love of the land. This is of utmost importance" (interview with Tali Freund, Israel Government Tourist Office, 2004). As a guide course coordinator summarized, "the guide is the ambassador of the state" (Haim Carel, interview, September 2004).

Many techniques, including Bible reading, use of maps, and panoramic outlooks, are common to Zionist Israeli hiking and Protestant pilgrimage, and indeed derive from shared historical sources (Feldman 2016, pp. 38–44). These commonalities provide the ground for Evangelical pilgrims' confidence in the Jewish guide, and lead them to assume that the guide shares their world-view. Yet there remains a gap between the understanding of the Bible as an Israeli national book of history or as a Christian book of divine revelation. These can be seen as two overlapping but conflicting faiths. Guides frequently gloss over this divide in the aim of creating a satisfying (and profitable) performance for their Christian clients, and the shared practices of Zionism and Protestantism (Feldman 2016, pp. 37–53; Ron and Feldman 2009) make this performance easy and 'natural' for Israeli guides. Sometimes, however, the differences may appear during the course of the tour.

In addition, guides of Christian pilgrim groups frequently come to see themselves not only as ambassadors for the country, but for Judaism as well. For many, the theological dispute between Christianity and Judaism over who is the heir to the Divine Promise, the history of Christian anti-Semitism, and the significance of the State of Israel as a site of Jewish power often lurk in the background. The haecceity—'this-ness'—of the landscape and the empowerment provided by a state in which Jews are the majority culture empower many guides to present Judaism as the root of Christianity. As one guide I interviewed explained, "every Christian is also a Jew, he just doesn't know it. But when he gets here, he discovers that" (Feldman 2016, pp. 123–26).

The vast majority of Christian pilgrims come to the Holy Land in order to reaffirm their faith in Christ, not to challenge it. The organization and scheduling of the tours are designed to minimize contact with locals and other narratives and focus on those faith-based sites that correspond to what pilgrims already know. However, the Jewish-Israeli guide is not only a facilitator of such faith (along with the pastor)[10], but a kind of grain of sand introduced into the sealed shell of the tour. While most guides will accommodate the beliefs of the group—who are, after all, his clients—the divergences in self-understanding occasionally surface, and present pilgrims with challenges to their faith. Moreover, the work demands emotional labor and empathetic competences (Hochschild 1983), requiring that he display enthusiasm even for sites and stories he does not identify with. This may result in role strains in some of the guiding performances and when shifting from on-stage guiding to off-stage personal life.

## 4. Jesus Wore Tefillin Like These Too: Bridging and Othering through Jewish Ritual

When I first began guiding Protestants in the Bible Land[11], nearly forty years ago, I would often read from the New Testament using intonations familiar to those of Evangelicals. Although I announced my Jewish identity, and took care not to refer to Jesus as "Messiah" or "our Lord and

---

[9] For a deeper discussion on the positionality of guides, between representation of place and personal integrity, see (Bunten 2015; Feldman and Skinner 2018).

[10] In the examples I chose for this article, the pastor plays no role. The groups' pastor is usually the tour leader, and is always responsible for leading prayers, administering communion, and preaching sermons, and should always be honored by the guide. Some pastors are extremely involved in providing explanations or pastoral care, while others have come to sit back and enjoy their vacation.

[11] While Catholic pilgrims usually refer to the land as the Holy Land, Evangelicals often take exception to this terminology, preferring to call it "Bible Land". Similarly, they may refer to themselves as Bible students rather than pilgrims—to distinguish themselves from Catholics and those who place emphasis on tradition and ritual.

Savior", I repeatedly found my position misconstrued. After a couple of days, participants would ask, "so when did you discover Jesus?" Theological explanations on Jewish messianic beliefs and historical expositions on the separation of Judaism and Christianity into two separate religions rarely made a difference; the same questions would continue, sometimes accompanied by pilgrims' testifying to me of their own beliefs in Jesus. For many Evangelicals, rhetoric, not ritual, is the primary vehicle of conversion. Listeners can undergo spiritual change by being receptive to speech that affirms Christian belief. By speaking of their beliefs in positive terms, I had shown them that I was on the road to accepting Christ. In Evangelical terminology, for many of them, if I was not born again (yet), by talking the talk, I had demonstrated that I had "come under conviction" (Harding 2000, pp. 59–60).

I looked for a way to mark my position as non-Christian without offending them. So, I tried ritual. On the day of our visits to the Western Wall, I packed my *tallit* (prayer shawl) and *tefillin* (phylacteries) in my backpack. When we arrived at the plaza in front of the wall, I wrapped myself in the tallit, recited the blessings in Hebrew and translated them into English, wound the tefillin around my arm, and placed them on my forehead. I then recited, in English, the Biblical text that served as the basis for the commandment: "You shall love the Lord your God with all your heart, with all your soul, and with all your might . . . And you shall bind [these words] as a sign upon your hand, and they shall be as frontlets between your eyes" (Deuteronomy 6, pp. 5, 8). I explained that as an observant Jew, Jesus would have worn tefillin like these for prayer[12]. As the group photographed me, while looking at the similarly attired worshippers at the wall, I sensed the coin dropping. "Ah, he's not one of us." Afterwards, they would ask me, "So what do *you Jews* think of Jesus?"

By citing these Old Testament verses at the Western (Wailing) Wall[13] (in Hebrew) at prayer time, while wearing tefillin, I met their expectations of the Israeli guide as "Biblical Hebrew". I had also succeeded in marking Judaism as other than Christianity through the ritual; this otherness commanded a certain respect from the pilgrims because it relied on verses of what was also Christian sacred scripture—Deuteronomy 5. They were also cited by Jesus as the foremost commandment in a widely known New Testament text (Mark 12, pp. 28–30). Even for Evangelicals, who often opposed others' traditions and rituals to their Scripture and faith, ritual did the trick; it marked the border[14]. Orthopraxy trumps orthodoxy here—even for Evangelicals.

After a year or so of these tallit and tefillin performances at the Western Wall, I reconsidered. At age 16, my father berated me for not putting on tefillin for prayer each morning. "Your father put on tefillin, your grandfather wore tefillin! Your cousins all wear tefillin! Only you—no. No good family!" I refused to wear the phylacteries to please my father. Was I now going to put them on for show to improve my position as guide—or to please the *goyim*?

Furthermore, was I not then saying—to them and ultimately to myself—that this is what *real* Jews do? How then would I explain, if asked, why I did not put on tefillin every day without appearing irreligious and irreverent? Because donning the tallit and tefillin and reciting the prayer could be seen as an act of commitment (which, in certain ways, it was), they might then expect me to behave as Orthodox do in order to be authentically Jewish in their eyes. This self-essentialization, especially given my decision *not* to wear tefillin each morning, also made me uncomfortable.

The insertion of the ritual into the guiding performance changes its nature both for myself and for the pilgrims. As the group's gaze moves from the leather straps on my arm to those worn by the

---

[12]   My explanation, although historically accurate, glosses over some differences. Tefillin of the period, as found at Qumran, were much smaller and less obtrusive than the leather boxes and straps worn by Orthodox men for daily prayer today. Jesus berates Pharisees who make "their phylacteries wide and the tassels on their garments long" (Matthew 23, p. 5) to display their piety to others.

[13]   The Western Wall has been visited by three Popes and numerous Christian leaders, who prayed and left notes at the site. Many Christian groups pray individually at the wall and place their prayer requests among its stones. It is a 'must' site for Evangelicals and many other Christians. This phenomenon awaits ethnographic research.

[14]   Whether Jewish ritual will continue to mark the border, given the tendency among some Evangelical congregations to adopt Jewish ritual (Dulin 2015; Shapiro 2019)—including the wearing of a *tallit*—remains to be seen.

Hasidim praying closer to the wall, what do they see? Do they reclassify me as an outsider? Have the Hasidim now become less strange? Or, perhaps, do the onlookers come to appreciate that 'their' Jesus wore straps much like these, and was, in fact, far more Jewish than they had imagined previously?

What is actually demonstrated through this ritual? Is the recital of the verses from Deuteronomy an affirmation of common faith? Does it perhaps show that what separates Jews and Christians is, first and foremost, a sense of belonging to a community sharing a common ritual?

This interaction is also shaped by the territory of the meeting. The performance of the rite at the Western Wall (and not, say, in a hotel conference room) marks me, as performer, as part of an ancient tradition (the Wall as remains of the Temple Mount), and as part of a larger community—the worshippers at the Wall. Through the reading of the Bible out loud there, "the Bible is transformed from a representation of memory into its actual physical recreation or embodiment. More than just an isolated memory, it becomes memory itself" (Mitchell 1997, p. 89). Thus, as the reader, I take part in the consecration of (Protestant Biblical) space, marking the Western Wall as the place of Jewish prayer. The pilgrims assume (as I often overheard) that it was so in Jesus' day too, even though it was then just a wall of the recently constructed Herodian Temple Mount, bordering a shopping street.

I do not assume, however, that this one-time ritual effects a deep change in pilgrims' religious categories.[15] One pastoral group leader, whom I guided several times on ten-day-long tours, including performing the ritual at the Western Wall, introduced me to a new group as "someone who knows the Bible, has a heart for the Lord and a heart for Jesus." If, for the pilgrim, Jesus is Lord, then a spiritual act that speaks to the Christian believer will likely be seen as an affirmation of Christian faith, in spite of my best efforts to mark the difference. It may be, however, that the performance of Scripture by someone who interprets the verses differently (wearing 'his' phylacteries) and the attractiveness of the Western Wall as a site of individual Christian prayer requests results in the acceptance of my ritual as an act of religious hospitality extended by the Jewish-Israeli native at 'his' home site, the Western Wall.

In conclusion, my performance of the rite for Evangelicals resulted in new reflection on the meanings of ritual, family loyalty, and instrumentalization for me. It might also lead pilgrims to consider the possibility of a different understanding and application of Scripture on the part of Jewish believers/practitioners. In both cases, the displacement of an 'internal' religious rite (wearing tefillin/reading Scripture) to an inter-religious context on shared sacred ground denaturalizes it and makes it an object of reflection.

## 5. You Serve the Same God: Confronting Evangelical Disgust with Religious Others

As mentioned above, Evangelical pilgrim itineraries are focused on Old and New Testament sites. In many of these Biblical sites, Protestants have the place for themselves: Catholics worship in the shrine, Protestants in the garden; Catholics at the historical church, Protestants at the archaeological site or nature view. Institutional Protestant pilgrimage in the 19th Century—especially Thomas Cook tours, were designed not only to make the holy sites more accessible, but to keep other pilgrims–especially the Russian Orthodox—at a distance (Ron and Feldman 2009; Lock 2003; Rogers 2003). Panoramic views from hilltops and enclosed bathing tents for immersion in the Jordan were two representative practices.

Nevertheless, there are some sites sacred to other groups visited by many Protestants. I have already discussed the Western Wall and the role played by the Jewish-Israeli guide there. Here, I will touch on the guide–pilgrim relationship at the Church of the Holy Sepulcher.

The Church of the Holy Sepulcher is the traditional site of Jesus' crucifixion and resurrection, venerated by Christian churches since the 4th Century. The site is shared by six Christian denominations, each with their own material signs: Altars, lamps, incense, clergy, and liturgy. The gold and silver

---

15　Kaell (2016) notes that Christian pilgrimage rarely 'fails' (or effects radical reversals) because it is framed by participants and churches as part of a larger Christian life-path, in which changes are incremental and understood in retrospect. "Ultimately", she writes, "the belief in *future meaning* is as important—perhaps more so—than immediate ritual success" (Kaell 2016, p. 393).

decorating the altars, the cacophony of voices in foreign languages, the vestments and beards of the Orthodox clergy, and the eddying crowds of mostly Catholic and Orthodox pilgrims all make Protestants, given their expressed theological divide between the material and the spiritual,[16] ill at ease. Some express outright revulsion. I often heard them complain: "Couldn't they just have left the tomb of Christ *alone*?" Nothing visible at the site corresponds with the image of the tomb and the hill of Calvary familiar to Protestants from their children's books, illustrated Bibles, and imaginations.

The incommensurability of church and imagination (Long 2003) was one reason leading the British Protestant General Gordon to invent an alternative site, the Garden Tomb, in the 19th Century, and fit it out to resemble an English rock garden.[17] Although some Evangelical groups avoid the church entirely, it is usually allotted some time on Evangelical itineraries as an archaeological and noteworthy historical site, though not for worship.

Some tour guides introduce the church by pointing out the Immovable Ladder on a window ledge above the entrance. The five-rung ladder was apparently placed there by the Armenian church, but on a ledge claimed by the Greek Orthodox. The right to remove the five-rung ladder was claimed both by the Greek Orthodox and the Armenian churches. The Status Quo Law governing the church declares that it must remain there—which it has since 1757 (see Cust 1929; Cohen 2008).

Such an introductory explanation has the effect of uniting Israeli guide and Protestant group in a sense of shared superiority over the squabbling, petty, and materialistic (in both senses of the word) Eastern Orthodox clergy[18].

As a guide, I would anticipate such reactions. I would first point out the Chapel of the Skull of Adam, underneath Golgotha. Then, pointing at the nearby Greek Orthodox mural, depicting the crucifixion, preparation for the tomb, and burial of Jesus, I focused their attention on the skull underneath the cross. I explained how the juxtaposition of skull and cross was a shorthand for a theology of Jesus as Second Adam, and the organic link between Adam—whose disobedience brought sin into the world by eating of the fruit of the Tree of Wisdom—and the cross as Tree of Life; through the cross and his sacrificial blood, Jesus, as Second Adam, atones for the sins of the first. "You see," I said, pointing to the Russian Orthodox prostrating themselves at the Stone of Unction, "you worship differently, but you serve the same God." Here, my Jewish-Israeli status positions me as an outsider to Orthodox and Catholic Christianity. The explanation, cast in a New Testament theology more or less familiar to them (1 Corinthians 15, pp. 45–49), meets with less resistance than if I were defending my own faith and practice. What is more, in order to encourage openness to other communities of worship, I strategically confirm their understanding that what unites Christians ('you' as opposed to 'me' or 'we') is, above all, faith in Jesus' atoning blood and his resurrection.

## 6. Is It Not Amazing How the Bible Is Always Right? Scripture, Tradition, and Science at Sacred Sites

An additional source of potential conflict arises around the veracity of archaeological and historic Christian sites. The demand for 'authentic' Biblical sites is certainly not limited to Protestants—it plays an important role in Catholic pilgrimage, classical Zionism, and contemporary Religious Zionism as well. For many Catholic and Orthodox pilgrims to the Holy Land, however, the sanction of

---

[16]　Houtman and Meyer characterize the Protestant view as follows: "The relation between religion and things has long been conceived ... (through) a set of oppositions that privilege spirit over matter, belief above ritual, content above form, mind above body, and inward contemplation above 'mere' outward action, producing an understanding of religion in terms ... of an interior spiritual experience" (Houtman and Meyer 2012, p. 1). Protestants are as prone to employ the material to convey religious devotion on pilgrimage as Catholics, though they employ different discourses and landscapes to do so (McDannell 1995; Ron and Feldman 2009). Protestant discourses, however, render their own material practices transparent. Denouncing the material marks Protestant borders vis-à-vis other Christian pilgrims.

[17]　For a succinct account of the 'revelation' and imperial politics round the Garden Tomb in the 19th Century, see (Monk 2002, pp. 34–36, 148–49).

[18]　This explanation is often used by Israeli guides leading Israeli school groups, and is an expression of their sense of the superiority of Judaism over Christianity (see Ramon et al. 2017, pp. 88–90).

ecclesiastical authorities who build churches and conduct liturgies at the site is more than sufficient legitimization—especially if the tradition there dates several centuries back. Not so for Protestants, who often set up a dichotomy between ('their') Tradition and ('our') Truth. On the other hand, they seek archaeological evidence to confirm Biblical truth. Indeed, the archaeological project in the Holy Land, beginning in the 19th Century, was spurred by the search for Biblical sites. The exploration of the Bible, the mapping and reconnaissance of the land for Western powers, and the development of Protestant pilgrimage have gone hand in hand since the 19th Century (Ben-Arieh 1979; Cohen-Hattab 2004; Monk 2002; Obenzinger 1999).

Nevertheless, the relationship between the Bible and archaeology is exceedingly complex. As Concannon summarizes in a recent survey article, although archaeology is committed to the scientific method, "archaeological materials are often consumed in biblical studies without due consideration or critical reflection. In some cases, this is even encouraged by archaeologists themselves, who may see benefits from such publicity" (Concannon 2013, p. 63). Moreover, for Evangelicals, the Bible is treated as an artifact—it provides the ultimate frame for interpretation of archaeological remains, but its authority cannot be challenged. As several pilgrims from different groups exclaimed to me, "isn't it amazing how archaeology always proves that the Bible is right!" In their interpretation of archaeological sites, guides are expected to provide proofs for Biblical stories, but not disproofs. As one Baptist minister said to me, "I didn't come all the way here to see where Jesus *wasn't*, I came to see where Jesus *was.*"

As, in most cases, archaeology can neither sufficiently confirm nor disprove Biblical accounts, this problem can often be overcome through selection of information—highlighting and contemporizing significant remains of Biblical times, while ignoring others—as is the practice among Israeli teacher–guides in general (see above). At many historical sites, what is important to archaeologists may be of negligible significance for Evangelical pilgrims. As one site manager writes about Megiddo, where Protestants ignore the massive Middle Bronze Age altar to look out over the valley they associate with the final battle of Armageddon (John 16), they "stand with their back to the archaeology" (quoted in Koren-Lawrence and Collins-Kreiner 2019, p. 144).

In some cases, however, scientific truth collides head-on with Evangelical certainties. Recently, I guided a delegation of three bus groups along with two other guides. At Tel Jericho, I was asked to explain the site to all three. I spoke of the Mount of Temptation (of Jesus), of the tight relations between Jericho and the Jerusalem Temple in Jesus' day, and pointed out the spring of Elisha referred to in the Book of Kings. I ended by saying that, unfortunately, archaeologists found no walls dating to the period of the Israelite conquest, the 12th CenturyBCE, as described in the Book of Joshua; I then turned the microphone over to one of my colleague guides. Without contradicting me outright, he hurriedly explained that the walls of the time might have been destroyed by later building. He then summoned up a theory mentioned in religious guidebooks and Evangelical websites, that the wall of that time fell down flat into the ground, as hinted in the Book of Joshua.[19]

At the Garden Tomb, the alternative Protestant site of Golgotha and the Tomb of Jesus (see above), groups are guided not by Israeli or Palestinian tour guides, but by Evangelical volunteers of the Garden Tomb Society. One of them, a veteran Australian volunteer, regularly delivered the following spiel: "Well, we can never be absolutely certain where Christ was buried. It could have been here (Garden

---

[19]  For an example of Christian archaeologists' confirmation of the Biblical account, see Bryan Wood's account in https://biblearchaeology.org/research/conquest-of-canaan/3625-the-walls-of-jericho. After arguing for an alternative dating for both the remains of Jericho and Joshua's conquest, he concludes, "If God did use an earthquake to accomplish His purposes that day, it was still a miracle since it happened at precisely the right moment". The consensus of scholars, however, supports archaeologist Kathleen Kenyon's claim that "the town walls of the Late Bronze Age, within which period the attack by the Israelites must fall by any dating, not a trace remains . . . The excavation of Jericho, therefore, has thrown no light on the walls of Jericho of which the destruction is so vividly described in the Book of Joshua" (Kenyon 1957, pp. 261–62). The scholarly discussion of the matter, however, is too detailed for most pilgrims—or for most guides. It is far easier to choose the hypothesis that will please the listeners.

Tomb), it could have been there (Holy Sepulcher), but the main thing is the tomb is empty! Praise the Lord." In confronting scholarly historical knowledge at Biblical sites, as in projects such as the Creation Museum in Kentucky (which combats evolution and advocates a young earth theory based on a literalist reading of the Bible), Evangelical discourse "both mobilize(s) th(e) authority (of science) and simultaneously deconstruct(s) it . . . (It) adopt(s) the symbolic form of science; but it also interrogates and destabilizes scientific claims to objectivity" (Butler 2010, p. 231). As Pastor Larry implied in his sermon (above), "You have heard it said: 'The water level has changed, the natural borders have changed, the rainfall has changed'; but I say: 'I never change'."

Unlike in the previous situations (Western Wall, Holy Sepulcher), my position as Jewish-Israeli guide endows me with few storytelling rights (Katriel 1997) for challenging Biblical truth based on archaeology. The critical scientific voice is familiar to Evangelicals from discussions at home, particularly on issues of Biblical inerrancy and evolution, and has engendered a series of well-established tactics for countering them and affirming Biblical inerrancy. Many of them employ the language of science to provide proofs for events described in the Bible, "simultaneously deploying and dismantling the truth effects of material objects" (Butler 2010, p. 243). As Ella Butler writes in her article on the Creation Museum, "the Bible becomes the historical record against which material evidence is tested. As the Bible is prior, in both a temporal and an epistemological sense, scientific research in the present can never disprove it" (Butler 2010, p. 241). Unlike in cases where I present alternative beliefs, interpretations of Scripture, or religious practices, when questioning the Biblical account, my scholarly voice becomes just one more voice in the chorus of unbelieving skeptics.

## 7. Deepening Faith/Encountering the Other

Finally, I suggest that the Jewish–Christian, guide–group encounter may offer a model for broadening religious faith based on encounters with the other and religious hospitality. Jerome Yehuda Gellman suggests that even for those who oppose relativism and accept the belief claims of their own religion as corresponding to the ultimate truth (as Evangelicals do), there is much profit to be gained by seeking what he refers to as telic truth:

> An aim of a religious practice has telic truth to the degree to which that aim is laudatory and to the degree to which that practice has what it takes to advance that aim, and ultimately to fulfil that aim. A religion can have several different practices and a plurality of aims, so I think more of telic truth *in* a religion than of the telic truth, per se, *of* a religion . . . (Gellman 2020, p. 31). Their telic truth may depart from the ultimate aims of my religion, and help me to better understand my own religious aims. And it might cause me to make adjustments to my own telic truths in considering it . . . (ibid, 32). So, religions other than mine do, at times, possess high levels of telic truth, while I do not embrace their core beliefs (ibid, 33).

Many Evangelicals have little occasion to witness the religious truths of others. When other religions are presented, they are often so subjugated to the soteriological framework of Christianity that the otherness of the religious other is effaced (Moyaert 2012), when it is not dismissed as outright idolatry.

The prominent role played by Messianic Judaism, actually a branch of Evangelical Christianity, in presenting Jewish symbols and rites to Evangelical publics intensifies this process. Many Evangelicals are averse to interreligious dialogue because they see it as a recognition of the validity of other systems of belief. One expression I often heard among pastors was: "There are many religions in this world. We don't have a religion. We have a personal relationship with Jesus Christ."

Those who do engage in interreligious dialogue often do so in formalized settings, with a representative of each religion ensconced behind his name tag and religious affiliation. There, they are expected to function as 'men of the cloth'—formal ambassadors of their religion—in which they serve as clerical experts. On Bible Land pilgrimages, on the other hand, the participants have not

arrived for the purpose of interreligious encounter; it involves people who are less schooled in the etiquette and diplomacy of official meetings, and often less educated in the basic principles, rituals, and beliefs of the other religions and of their own. For most Evangelicals, the encounter with a Judaism not predigested by Messianics is new. Moreover, the meeting occurs in an unfamiliar power situation—not in a church, but in Israel, a place where the pilgrims are guests, and the Jewish guide can play the host. This hospitality has a dynamic of its own: Ritual hospitality, especially in public space, proclaims solidarity—not by fixing theological truths, but by engaging in an ongoing process of encounter and mutual recognition (Moyaert 2011, p. 26). This dynamic often fosters a generosity of spirit (Siddiqui 2015, p. 1) in the interpretation and reception of particularistic and potentially competitive messages.

Within this environment of religious hospitality, the guide/host occupies a liminal position—as native to the Land of Israel, part of the Chosen People, but not (for many Evangelicals—not yet) a Christian, he has a certain authority to challenge pilgrims' faith. As Steve, a veteran guide for such groups, explained:

> I make it a point to ask them how they feel about me, as a Jew who does not believe in Jesus, reading to them from their holy book, but the fact is, as I tell them, it is a book that I take very seriously in my searching, and in that spirit I read from it to you. If I were a Christian I couldn't tell them all the things I do. Because then I would be part of the club. Not all participants appreciate it; some would rather be soothed and confirmed in the faith they brought in to the trip. Some would rather have less explanation and questioning and more time to meditate in silence. [I tell them that] in Judaism, the Messiah is someone who brings salvation, redemption. Jesus was here and the world hasn't improved much. Sometimes I would feel their adrenalin mounting, and I said—'good! That's the way it was 2000 years ago, when Jesus was here in the synagogue, and some walked out of the synagogue. And now it's a good time to eat St. Peter's Fish'. Who in their surroundings talks with them in that way?"

For the Jewish-Israeli guides, even if they are at home in the landscape, the aesthetics and music of Christianity and the mutual care displayed by the pilgrims they guide may seduce them (Feldman 2015). Often, as I can attest, guides get 'caught up' in performance. The ways that the performance aroused my own religious reflection at the Western Wall indicate that it was more than a theatrical mimicry of a religious rite. Likewise, as some post-voyage letters of pilgrims attest, the encounter with the Jewish-Israeli guide, especially *because* it is not hard-wired into the itinerary, may lead to a new appreciation of Judaism as another path to God. As Edward Schieffelin writes: "We are, in effect, more performative than we intend, and we are in good measure 'submitted to' our performativity as part of our active being-in-the-world" (Schieffelin 1998, p. 197). Performance does not merely *express* feelings, intentions, or beliefs; it helps *create* them.

## 8. Conclusions

I have outlined several real or potential tensions in contemporary Holy Land pilgrimage: Between Christian faith and Israeli national heritage, between Evangelical Christianity and Judaism, between Evangelical faith and practice and those of other Christian groups at holy sites, and between faith-based certainties and scientific skepticism. In each of these cases, the Israeli tour guide can play a mediating role, smoothing over the potential conflicts, or on the contrary, highlighting differences and tensions.

The potential for conflict in each of these issues varies, as does the authority that the guide can claim in speaking of them. In the case of Christian faith versus Israeli heritage, the practices of Bible reading, collapsing of time-frames (Katz 1985), and pointing from panoramas (Feldman 2016, pp. 43–51) are so historically intertwined and so similar that Israeli practice is understood as confirming Protestant religious experience. This performance, in turn, affirms the Evangelical appropriation of the Bible, as pilgrims assume that a sympathetic portrayal of Christianity is a declaration of allegiance of one who "knows the Bible and knows the Lord". The pilgrims' expression of this view can create discomfort

on the part of the Jewish guide, and the use of ritual as border marker is one strategy for asserting personal integrity without alienating the pilgrims-as-clients. In explaining other Christian groups and practices, often foreign to Evangelicals, strategic evocation of New Testament passages may open a window towards comprehension or, at least, compassion. In each of these cases, the guide negotiates between his (national, religious, humanist) commitments and his interest in accommodating the beliefs and expectations of the pilgrims, who are his clients. He does this through selection of appropriate information or display of understanding and empathy, even for views he does not share. In each of these cases, the guide's position as native and host endows him with storytelling rights, especially if parts of the story/practice are familiar and even sacred.

Not so in the case of a conflict between scientific truth and Biblical certainties. Undoubtedly, the guide may avoid conflict by suppressing scientific evidence wherever it clashes with Evangelical belief, while citing evidence that supports it. However, in case of outright contradictions (was there a wall of Jericho at the time of the Israelite conquest? Is the Garden Tomb 'authentic'?), the collision may be hard to escape. Here, the guide gets no 'points' as native, and the skeptical, scholarly voice may arouse hostility.

Finally, I suggest that Evangelical pilgrims and Jewish–Israeli guides are not inevitably locked into positions determined once and for all by an ascribed or long-practiced identity. Travel is, as Judith Adler terms it, a "performed art": "Travel lends itself to dramatic play with the boundaries of selfhood, and the character ideals of the performers and their audiences are as various as the performances . . . Enduring identities are often narratively constructed on the basis of brief adventures" (Adler 1989, p. 1385). In Christian pilgrimages to the Holy Land, the vast majority of Christians travel in order to reaffirm their faith in Christ, not challenge it. Guides do their work to earn a living.[20] Yet the interactions that take place—some deliberate, some out of the control of the performers—demonstrate the potential of faith encounters though pilgrimage not only to mark borders, but to discover a shared humanity that transcends them.

**Funding:** Research was partially funded by Israel Science Foundation Grant 291/13 (with Prof. Yvonne Friedman) and by a Chateaubriand scholarship from the French Embassy in Tel Aviv, Israel, for research at Aix-Marseille University, Marseille, France (2020).

**Conflicts of Interest:** There has been no conflict of interest in researching or writing this article.

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
