# Peer review of "How Can You Know the Bible and Not Believe in Our Lord? Guiding Pilgrims across the Jewish–Christian Divide"

_religions, doi:10.3390/rel11060294_

Round 1

Reviewer 1 Report

No comments. 

Author Response

Thank you very much.

Reviewer 2 Report

A fascinating and well-written article and one which I would seriously consider using in my own seminar on Israel. The auto-ethnographic approach is a good starting point for presenting an array of issues that can be examined in  subsequent research: one that comes to mind is contrasting and comparing differences in responses of the different Christian denominations guided by the author as well as those of Jewish tourists. This I think will allow this researcher and others to delve more deeply into the relevant theoretical concerns delineated in the proliferating literature on the anthropology of tourism.

Author Response

Thank you very much.

Reviewer 3 Report

This was a really fascinating read. The author clearly has extensive experience with Evangelical Christian pilgrims and their interests in Holy Land pilgrimage as well as many of the resulting tensions with both Israeli national identity and Jewish belonging. The ethnographic examples contained herein are especially compelling and, I think, demonstrate many of the ideological divides the author is primarily concerned with.

My overall critique is therefore to highlight the general structure of the article. While it is clearly written in the auto-ethnographic milieu of Geertzian Reflexive Anthropology, there is simply so much going on that it is a little difficult to follow the main threads throughout the work. There's the division between Jewish identity and Christian presumption/conversion, between affirmation of faith and challenge, between rhetoric and ritual, between pilgrim experience and tourist agency framing, between Israeli identity and Euro-American nationalism, orthodoxy versus orthopraxy, and between science and religion.

The introduction does a fine job of framing the methods and positionality, for example (for which the author should be lauded), but there were several sections where I wasn't entirely clear on how each part was intended to work cohesively towards a unifying analytical point. In other words, the article reads very reflexively (for obvious reasons) but the main argument, the "what does all this tell us?," is often unclear. I've noted in the blinded manuscript, by line, where I see this most acutely if that is more helpful. In short, by bringing each dichotomy together under a primary framing device, I think this article could quickly go from good to great and will then rely less on the open-ended questions presented in the abstract and in the introduction and more on the conclusion as woven throughout.

Author Response

I appreciate the comments for revision of the third reader, and have gone through the article, making the following changes:

  1. I changed the questions in the abstract and introduction so that they correspond better with the subtitles and sections of the article and the conclusions, deleting general questions not sufficiently treated in the article.
  2. I reduced references to Orientalism; although I think that nationalist and colonial views are present in the pilgrimage encounter, I cannot expound the full problematic in the course of this article. I hope to treat this elsewhere. I added an additional footnote and several bibliographical references on the subject.
  3. I clarified the passages that were unclear and removed the final postscript paragraph.